# Revisiting Genetic Epidemiology with a Refined Targeted Gene Panel for Hereditary Hearing Impairment in the Taiwanese Population

**DOI:** 10.3390/genes14040880

**Published:** 2023-04-07

**Authors:** Yen-Hui Lee, Cheng-Yu Tsai, Yue-Sheng Lu, Pei-Hsuan Lin, Yu-Ting Chiang, Ting-Hua Yang, Jacob Shu-Jui Hsu, Chuan-Jen Hsu, Pei-Lung Chen, Tien-Chen Liu, Chen-Chi Wu

**Affiliations:** 1Department of Otolaryngology, National Taiwan University Hospital, Taipei 10002, Taiwan; k32650805@gmail.com (Y.-H.L.);; 2Graduate Institute of Medical Genomics and Proteomics, National Taiwan University College of Medicine, Taipei 10055, Taiwanpaylong@ntu.edu.tw (P.-L.C.); 3Department of Otolaryngology, Buddhist Tzuchi General Hospital, Taichung Branch, Taichung 42743, Taiwan; 4Department of Medical Genetics, National Taiwan University Hospital, Taipei 10041, Taiwan; 5Graduate Institute of Clinical Medicine, National Taiwan University College of Medicine, Taipei 10002, Taiwan; 6Department of Otolaryngology, National Taiwan University College of Medicine, Taipei 10002, Taiwan; 7Department of Medical Research, National Taiwan University Hospital Hsin-Chu Branch, Hsinchu 30261, Taiwan

**Keywords:** hearing loss, sensorineural, deafness, genetic testing, molecular epidemiology, high-throughput nucleotide sequencing, genetic counselling

## Abstract

Hearing impairment is one of the most common sensory disorders in children, and targeted next-generation sequencing (NGS)-based genetic examinations can assist in its prognostication and management. In 2020, we developed a simplified 30-gene NGS panel from the original 214-gene NGS version based on Taiwanese genetic epidemiology data to increase the accessibility of NGS-based examinations. In this study, we evaluated the diagnostic performance of the 30-gene NGS panel and compared it with that of the original 214-gene NGS panel in patient subgroups with different clinical features. Data on the clinical features, genetic etiologies, audiological profiles, and outcomes were collected from 350 patients who underwent NGS-based genetic examinations for idiopathic bilateral sensorineural hearing impairment between 2020 and 2022. The overall diagnostic yield was 52%, with slight differences in genetic etiology between patients with different degrees of hearing impairment and ages of onset. No significant difference was found in the diagnostic yields between the two panels, regardless of clinical features, except for a lower detection rate of the 30-gene panel in the late-onset group. For patients with negative genetic results, where the causative variant is undetectable on current NGS-based methods, part of the negative results may be due to genes not covered by the panel or yet to be identified. In such cases, the hearing prognosis varies and may decline over time, necessitating appropriate follow-up and consultation. In conclusion, genetic etiologies can serve as references for refining targeted NGS panels with satisfactory diagnostic performance.

## 1. Introduction

Hearing impairment is one of the most common sensory disorders in children. The estimated prevalence is 1.9 out of 1000 live births, which increases to approximately 2.7 per 1000 before the age of 5 years and 3.5 per 1000 during adolescence [1]. Hereditary hearing impairment (HHI), the cause being attributed to pathological genetic variants, accounts for 50% of sensorineural hearing impairments (SNHI) in children [2,3,4,5]. To date, more than 200 genes have been implicated in HHI (http://deafnessvariationdatabase.org/, last accessed on 19 February 2023). Because of its impact on speech/language development and cognitive functions [6,7], it is crucial to intervene in HHI early in life, and genetic testing assists in its prognostication and selection of management [8,9,10,11,12].

In our previous study, we investigated the genetic epidemiology of Taiwanese patients with idiopathic SNHI. Common deafness-associated genes, in order of prevalence, included *GJB2*, *SLC26A4*, *OTOF*, *MYO15A*, and *MT-RNR1*, which are similar to those found in other populations [13]. However, we did not investigate the genetic underpinnings of the patient subgroups based on clinical features. Usami et al. reported different causative genes among different age groups using data from a nation-wide cohort of 10,047 subjects [14]. Liao et al. found that the genetic diagnosis was associated with age at testing, ethnicity, onset of SNHI, as well as the uni- or bilaterality of SNHI, and demonstrated that 29.4% of the diagnoses correlated well with the prognoses [15]. Accordingly, clinical features also provide useful information for interpreting genetic results and achieving a diagnosis.

With advances in sequencing technologies, next-generation sequencing (NGS)-based genetic examinations have become first-tier diagnostic tests for SNHI in many laboratories. We previously implemented a 214-gene NGS panel in clinical practice, which led to a remarkable increase in the diagnostic rate—from 25% using conventional Sanger sequencing to approximately 50% using NGS [13]. The design of commercial NGS diagnostic panels for deafness differs across laboratories, with the number of sequenced genes ranging from 12 to 372 [16,17]. The difference in the number of sequenced genes most likely represents a trade-off between diagnostic yield and the cost of genetic examinations. As genetic examinations for childhood SNHI are not reimbursed by the National Health Insurance in Taiwan, we further developed a simplified 30-gene panel in 2019 based on Taiwanese epidemiological data to increase the accessibility of NGS-based examinations to patients [13]. However, the diagnostic performance of the simplified NGS panel remains unclear.

In this study, we compared the diagnostic yield of the 30-gene NGS panel with that of the original 214-gene NGS panel in patient subgroups with different clinical features. Our results demonstrated that, except for patients with late-onset SNHI, the simplified NGS panel retained a diagnostic performance comparable to that of the 214-gene NGS panel.

## 2. Materials and Methods

### 2.1. Recruitment of Patients

We enrolled patients who underwent NGS-based genetic examinations for idiopathic bilateral SNHI at the Department of Otolaryngology, National Taiwan University Hospital, between January 2020 and December 2022. Patients with coexisting acquired risk factors for childhood SNHI, including cytomegalovirus (CMV) infection, premature birth, meningitis, neonatal icterus, or previous exposure to noise or ototoxic medications, were excluded. In total, 350 patients were included in the analysis, and all participants self-identified as Han Chinese.

This study was approved by the Institutional Review Board of the National Taiwan University Hospital (IRB: 201803092RINB) and written informed consent was obtained from all participants, their parents, or guardians. Data reporting followed the Strengthening the Reporting of Observational Studies in Epidemiology (STROBE) guidelines.

### 2.2. Clinical Features of the Study Cohort

All patients underwent comprehensive history taking, physical examinations, and a battery of audiological examinations. The age of onset of hearing loss was collected for each patient and they were divided into five groups: prelingual (below age 4 years), preschool (aged between 4 and 6 years), early school to teenage (aged between 6 and 18 years), young adult (aged between 18 and 40 years), and late-onset (aged > 40 years) [14,18,19,20]. The battery of audiological examinations included tympanometry, sound field audiometry, pure-tone audiometry, otoacoustic emission tests, auditory brainstem response tests, and auditory steady-state response tests. Audiological examinations were performed based on the age and neurological status of the subjects [21]. The hearing level of the better ear was determined by averaging the thresholds at four frequencies (500, 1000, 2000, and 4000 Hz), and was classified as normal (<20 dB), mild (21–40 dB), moderate (41–70 dB), severe (71–95 dB), or profound (>95 dB) hearing loss [22]. Progression of hearing loss was defined as a greater than 15 dB increase in hearing level between two consecutive tests [15].

### 2.3. Next-Generation Sequencing (NGS)-Based Genetic Examinations

Since January 2020, we have provided two versions of targeted NGS panels, the original 214-gene panel and the simplified 30-gene panel, in our clinical practice. The complete lists of genes are summarized in Table 1, and the NCBI reference sequences of the 30-gene panel are listed in Appendix A. The simplified 30-gene panel did not include *TMPRSS3*, *LOXHD*, and *CDH23* because of relatively low prevalence of the hereditary hearing loss caused by these genes [13].

The details of the NGS-based genetic examinations, including sample preparation, DNA sequencing, and data analyses, were in line with our standard operating procedures and have been described elsewhere [23]. Briefly, DNA samples extracted from peripheral blood mononuclear cells were sequenced using Illumina MiSeq (Illumina Inc., San Diego, CA, USA), which generated paired-end reads of 300 nucleotides with a mean read depth of 150x. Genomic data processing and variant calling complied with the Genome Analysis Toolkit (GATK) best-practice workflow designed by the Broad Institute, with BWA-MEM used for mapping reads to the reference genome (GRCh37/hg19), Picard for duplicates marking, and GATK HaplotypeCaller for variant calling [24,25,26].

Identified variants, including single-nucleotide polymorphisms (SNPs), small insertions, and small deletions, were annotated using ANNOVAR [27]. In addition to the effect of variants on protein coding delineated via their location relative to the coding region, allele frequencies from the 1000 Genomes Project [28] and Genome Aggregation Database (gnomAD) [29] were also annotated. In silico prediction scores including SIFT and PolyPhen-2 were also included to determine the pathogenicity of the nonsynonymous variants [30,31].

With reference to the Deafness Variation Database and ClinVar [32,33], all variants were then categorized according to the American College of Medical Genetics and Genomics (ACMG) guidelines [34]. Genetic diagnosis was achieved when one pathogenic/likely pathogenic, dominant variant or bi-allelic pathogenic/likely pathogenic, a recessive variant, was detected. Sanger sequencing was performed to confirm the identified variants.

### 2.4. Statistical Analysis

The chi-square test was used to compare categorical variables, and Fisher’s exact test was used if any of the expected cell counts < 5. The effect size was expressed using odds ratios (ORs) with 95% confidence intervals (CIs). Linear regression analysis was used to analyze the relationship between the hearing deterioration and the follow-up time. A *P* value of less than 0.05 was considered statistically significant. All statistical analyses were performed using statistical software R (version 4.2.2).

## 3. Results

### 3.1. Clinical Features of the Study Cohort

The participant demographics and audiological profiles are summarized in Table 2. The mean age was 12.4 ± 13.8 years, and the study cohort comprised 171 (49%) female and 179 (51%) male patients. SNHI manifested at the prelingual age in 49% patients, followed by preschool (20%), young adult (14%), early school to teenage (12%), and late-onset (5%). Of 350 patients, 234 (67%) were classified as mild-to-moderate SNHI, while the other 116 (33%) patients had severe-to-profound SNHI at the first visit.

### 3.2. Distributions of Genetic Etiologies by Severity of SNHI and Age of Onset

Figure 1 shows the distribution of genetic etiologies by the severity of SNHI at diagnosis. Details of causative variants on relevant genes are summarized in Appendix A. The *GJB2* variants p.V37I and p.M34T were classified as pathogenic, as in our previous report. Both panels covered the non-coding regions of exon 1 of GJB2; however, no pathogenic variant was detected. The most common genetic causes in patients with mild-to-moderate SNHI were *GJB2* variants (39%), followed by *SLC26A4* (4%)*, MYO15A* (1%), and *OTOF* (1%) variants. By contrast, *SLC26A4* (17%), *MYO15A* (8%), and *OTOF* (3%) variants were more prevalent in patients with severe-to-profound SNHI, and the percentage of *GJB2* variants was relatively lower in this group (14%). The rates of negative results on genetic tests were similar regardless of the severity of SNHI (50% for mild-to-moderate SNHI vs. 46% for severe-to-profound SNHI).

The distribution of genetic etiologies by the age of onset is shown in Figure 2. The rate of negative result tended to increase with age of onset, with 61% in the late-onset group and only 46% in the prelingual-onset group. *GJB2* variants were more commonly identified in patients with prelingual-onset and preschool children, accounting for 35% and 36%, respectively, compared with 24%, 21%, and 17% in patients with onset at ages of early school to teenage, young adults, and older than 40 years old, respectively. In contrast, the percentage of *SLC26A4* variants was relatively higher in patients from early school to young adulthood (12% and 15%, respectively) than the other age groups (6% for prelingual onset group, 7% for preschool onset group, and 6% for the late-onset group).

### 3.3. Comparison of Diagnostic Rates between Different Targeted NGS Examinations

The clinical features of patients tested with different panels are summarized in Table 3. There was a total of 271 patients tested with a 30-gene panel and 79 patients tested with the 214-gene panel. The two groups did not differ significantly in terms of age (12.0 ± 12.3 vs. 13.8 ± 16.2) or gender (137 females and 134 males vs. 34 females and 45 males). The age of onset was also similar between the two groups, while there was significant difference in the severity of hearing loss, with a higher ratio of severe-to-profound hearing loss in the 214-gene panel group (54%) compared to the 30-gene panel group (27%).

The diagnostic rates for 30-gene and 214-gene panels are summarized in Table 4. The overall diagnostic rate of the 30-gene panel was slightly lower than that of the 214-gene panel (51% vs. 56%), while the difference was not statistically significant. In addition, the diagnostic rates of the two panels were similar in the patients with mild-to-moderate SNHI (51% vs. 53%) and those with severe-to-profound SNHI (52% vs. 58%). However, when the patients were sub-grouped based on the age of onset, the diagnostic rate was significantly lower in the 30-gene panel than in the 214-gene panel in the late-onset group (17% vs. 83%; *p* = 0.01) but not in the other four groups.

### 3.4. Prognosis of Hearing in Patients with Different Genetic Diagnoses

The prognoses of hearing in patients with negative results and confirmed genetic diagnosis are summarized in Table 5. The rate of progressive or profound SNHI was as high as 60% in patients with negative results, which was significantly higher than patients with *GJB2* variants (41%; OR: 0.48; CI: 0.25–0.92; *p* = 0.03) but lower than patients with *MYO15A* variants (100%; OR: 14.9; CI: 0.85–260.5; *p* = 0.007). Figure 3 shows the longitudinal changes in the hearing levels of patients with negative results. The prognosis was highly variable, but a deteriorative tendency was still observed (y = −1.5x −1.4, where x and y denote time in years and change in hearing level in dB, respectively, *p* = 0.006).

## 4. Discussion

In this study, the overall diagnostic yield of targeted NGS examinations was 52%, with slight differences in the distributions of genetic etiologies between patients with different degrees of SNHI and ages of onset. The diagnostic rates of the 30-gene and 214-gene panels were similar, regardless of clinical features, except that the 30-gene panel had a lower detection rate in patients with late-onset SNHI. The prognosis of hearing loss in patients with negative genetic testing results is highly variable and some might decline over time.

The overall diagnostic rate of 52% is consistent with our previous report [13]. The results of previous studies were highly variable and ranged from 15% to 79% [14], which may reflect ethnic differences, selection of patients or different designs of targeted NGS panels. Yan et al. reported the lowest detection rate of 15% via applying a 180-gene NGS panel universally to a multi-ethnic cohort, where specific detection rates varied from 0% to 57% based on ethnicity [35]. A systemic review, including 1355 populations of different ethnicities from 1165 studies, indicated that over 90% of the subjects were of European and Asian ancestry. This may affect the interpretation of the detected variants and lower the diagnostic yields in other underrepresented populations, including those of Latino, American, African, and indigenous North American ancestry [36]. Furthermore, the diagnostic rate was also relatively lower when NGS served as a second-tier approach in *GJB2*-negative patients. For instance, Brozkova et al. reported a diagnostic rate of 21% in the Czech population [37,38]. The highest detection rate of 79% was reported by Budde et al., who used two-tiered approaches with 12-gene and 94-gene NGS panels for Egyptian patients, with diagnostic yields of 36% and 75%, respectively [16]. The difference in diagnostic yields also suggests that reducing the number of target genes might lead to a decrease in detection rates. Therefore, it is crucial to strike a balance between the examinations cost and diagnostic yield.

Nevertheless, narrowing down the targeted regions is a logical choice to modify genetic testing for clinical use, because most panels detect pathogenic variants in much fewer responsible genes than expected [14]. Additionally, whole-exome sequencing (WES) only detected causative variants in less than 40 genes, and the diagnostic rates were also around 50% [39,40], implying that diagnostic rates may not improve to the same extent as the targeted coding regions expand. The present study revealed that the diagnostic rates of 30-gene and 214-gene panels in Taiwanese patients were comparable, suggesting that the simplified 30-gene version does not compromise the detection rate. Therefore, 30-gene targeted sequencing can be adopted as a first-tier genetic test for Taiwanese patients with suspected hereditary hearing loss.

Notably, the 30-gene panel showed a lower detection rate in the late-onset group than the 214-gene panel. A possible explanation for this is that we developed the 30-gene panel on the basis of our previous epidemiological studies, in which the majority of the study population was aged less than 40 years [13]. It has been reported that late-onset SNHI might be associated with more diverse genetic causes [41,42]. Owing to the complex genetic underpinnings, a more comprehensive NGS may be warranted to address late-onset SNHI.

In the present study, the responsible genes were slightly different between patients with different degrees of SNHI, with *GJB2* variants more commonly associated with mild-to-moderate SNHI, and *SLC26A4*, *MYO15A*, and *OTOF* variants more commonly associated with severe-to-profound SNHI. These findings were consistent with the previously reported phenotype–genotype correlations [9]. On the other hand, the responsible genes by age of onset might also reflect the clinical course of SNHI. The higher percentage of *SLC26A4* variants in early school to adult patients was compatible with the associated fluctuating and progressive SNHI. It should be noted that approximately 50% of these patients pass neonatal hearing screens and have an average hearing level of up to 80 dBHL later in their life [43].

In cases with negative results, more comprehensive genetic examinations, such as whole-genome sequencing and/or long-read sequencing can be performed whenever clinically indicated. Although structural variants calling mostly requires whole-exome/genome data, recently, several novel bioinformatic tools have been proposed to detect copy number variation (CNV) using NGS panel data. The CNV calling algorithm can be based on read coverage, paired end, split read and assembly [44], while most tools that use panel data are based on read coverage [45]. Because the accuracy of the tools varies depending on the dataset, a combination of these tools is generally suggested for CNV analysis [46]. Our preliminary results using CODEX2 [47,48], DeCON [49] and CNV [50] indicate the promising clinical usage of these tools, as *STRC* homodeletions were detected among five patients with negative genetic testing results, and all of them had mild-to-moderate hearing loss, which is compatible with the typical phenotypes of STRC mutations. However, owing to the high false-positive rates of these tools, the genetic diagnosis of CNV should be achieved with confirmatory tools, namely multiplex ligation-dependent probe amplification, long-ranged polymerase chain reaction, or long-read sequencing, which enables haplotype phasing [44].

Despite efforts to diagnose the responsible genes using high-throughput data, a common dilemma faced by clinicians is to counsel patients with negative genetic testing results. Several predictive models have been proposed to depict the clinical course of SNHI related to certain deafness genes. Chen et al. used multivariate generalized estimating equations analyses on 707 audiological records of 227 patients and concluded that hearing levels in patients with *GJB2* mutations could be estimated using the following formula: predicted hearing level in dBHL = 3.78 + 0.96 x baseline hearing level in dBHL + 0.55 x duration of follow-up in year [9]. For patients with *MYO15A* mutations, an average progression rate for non-profound SNHI was 6.3 ± 4.8 dBHL per year [10]. In contrast, there is a paucity of studies that delineate the prognosis of hearing-impaired patients with negative genetic results. In the present study, although the prognosis of hearing in patients with negative genetic results was highly variable, a tendency for deterioration was still observed: 60% of the patients exhibited progressive or profound SNHI at diagnosis. The percentage was between that of patients with *GJB2* and *MYO15A* variants, and the average deterioration of SNHI in patients with negative genetic results was estimated to be 1.5 dBHL deterioration per year (Figure 3). This information may be useful in counselling patients with negative genetic results regarding hearing outcomes.

This study had several limitations that merit further discussion. First, the study was retrospective, and patients may have selected different NGS panels based on economic considerations. This might have led to a recruitment bias, particularly in the context of different socioeconomic statuses between the participants. Second, we excluded patients with coexisting acquired risk factors for childhood SNHI to ensure the diagnosis of HHI, while the strict criteria may result in minor selection bias and precludes an analysis of interactions between genetic and environmental factors. Third, structural variants (including CNV) and mosaic variants were not analyzed owing to the technical constraints of the short-read NGS; thus, it is possible that some pathogenic variants might have been missed, which requires further investigation.

## 5. Conclusions

The genetic etiologies vary between patients with different degrees of SNHI and ages of onset and could serve as references for refining targeted NGS panels. When appropriately designed, simplified NGS panels targeting the predominant deafness genes can retain satisfactory diagnostic performance. Although highly variable, hearing deterioration is not uncommon in patients with negative genetic results, and appropriate counselling should be provided accordingly.

## Figures and Tables

**Figure 1 genes-14-00880-f001:**
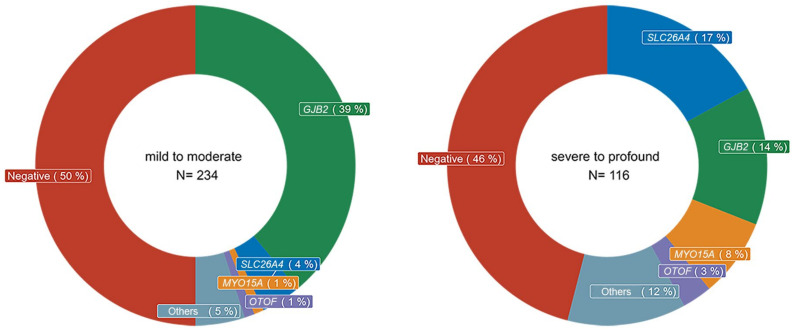
Genetic etiologies by severity of sensorineural hearing impairment (SNHI). The most commonly affected gene was *GJB2* in patients with mild-to-moderate SNHI, while *SLC26A4*, *MYO15A*, and *OTOF* were more commonly associated with severe-to-profound SNHI. The rates of negative results for genetic testing were similar regardless of the severity of SNHI.

**Figure 2 genes-14-00880-f002:**
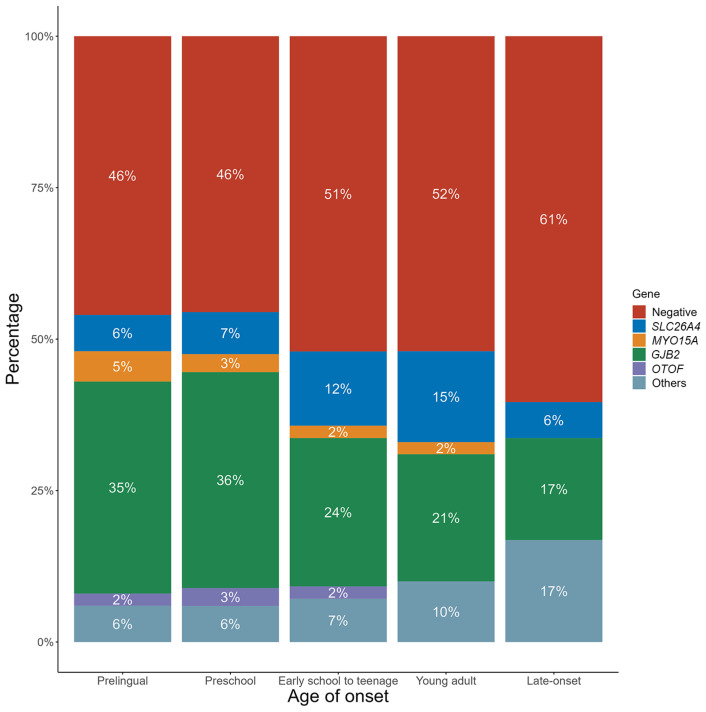
Genetic etiologies by age of onset. The rates of negative results increased by age, with *GJB2* variants being more likely detected in patients with onset at prelingual age. By contrast, the percentage of *SLC26A4* variants were relatively higher in patients with onset from early school to young adult than the other age groups.

**Figure 3 genes-14-00880-f003:**
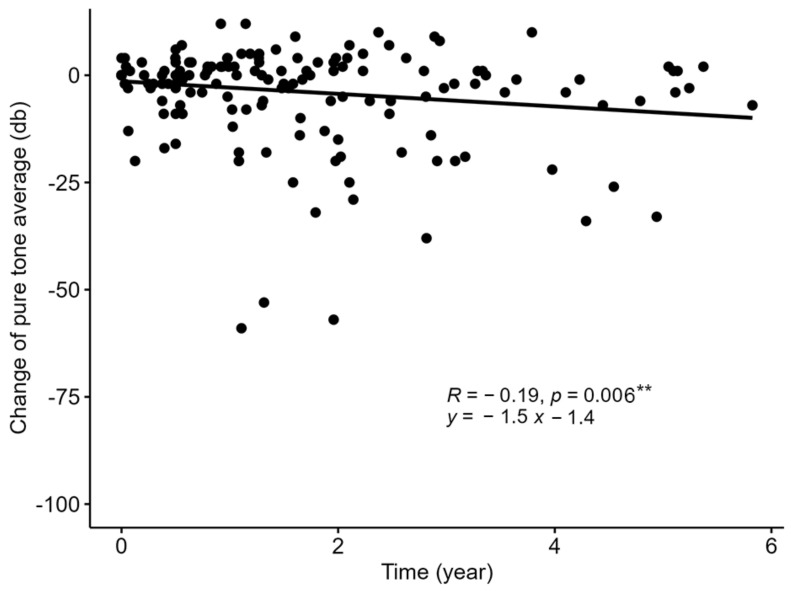
The sequential changes in hearing level in patients with negative genetic testing results. The prognosis for patients with negative genetic testing results was highly variable but a deteriorative tendency was still observed. ** *p* < 0.01.

**Table 1 genes-14-00880-t001:** Comparison of the 30-gene and 214-gene panels.

	Study Cohort (N = 350)
30-gene panel	*AIFM1, DIAPH3, EDN3, EDNRB, EYA1, FOXI1, GJA1, GJB1, GJB2, GJB3, GJB4, GJB6, KCNJ10, KCNQ4, MITF, MYO15A, OTOF, PAX3, PCDH9, PJVK, POU3F4, POU4F3, SIX1, SIX5, SLC26A4, SNAI2, SOX10, STRC, MT-RNR1, MT-TL1*
214-gene panel	*ACTB, ACTG1, ADCY1, ADGRV1, AIFM1, ALMS1, ATP2B2, ATP5MF, ATP6V0A4, ATP6V1B1, ATP6V1B2, BCS1L, BDP1, BSND, BTD, CA2, CABP2, CACNA1D, CATSPER2, CCDC50, CD164, CDC14A, CDH23, CEACAM16, CEMIP, CEP78, CHD7, CIB2, CISD2, CLDN14, CLIC5, CLPP, CLRN1, COCH, COL11A1, COL11A2, COL2A1, COL4A3, COL4A4, COL4A5, COL4A6, COL9A1, COL9A2, COL9A3, CRYL1, CRYM, DCDC2, DIABLO, DIAPH1, DIAPH3, DMXL2, DSPP, DUOX2, ECE1, EDN3, EDNRA, EDNRB, ELMOD3, EPS8, EPS8L2, ERAL1, ERCC2, ERCC3, ESPN, ESRRB, EYA1, EYA4, FAS, FGF3, FGFR1, FGFR2, FGFR3, FITM2, FOXI1, GATA3, GIPC3, GJA1, GJB1, GJB2, GJB3, GJB4, GJB6, GPSM2, GRHL2, GRXCR1, GRXCR2, GSDME, GSTP1, HAL, HARS1, HARS2, HECTD3, HGF, HOMER2, HSD17B4, ILDR1, JAG1, KARS1, KCNE1, KCNJ10, KCNJ11, KCNQ1, KCNQ4, KITLG, LARS2, LHFPL5, LHX3, LOXHD1, LOXL3, LRP5, LRTOMT, MARVELD2, MCM2, MET, MITF, MPZ, MPZL2, MSRB3, MTAP, MYH14, MYH9, MYO15A, MYO1C, MYO1F, MYO3A, MYO6, MYO7A, NARS2, NDP, NF2, NLRP3, NR2F1, OPA1, OSBPL2, OTOA, OTOF, OTOG, OTOGL, OTOR, P2RX2, PAX3, PCDH15, PCDH9, PDZD7, PEX1, PEX6, PEX7, PHYH, PJVK, PMP22, PNPT1, POLR1C, POLR1D, POU3F4, POU4F3, PRPS1, PTPRQ, PTRH2, RDX, RIPOR2, ROR1, S1PR2, SCARB2, SEMA3E, SERPINB6, SIX1, SIX5, SLC17A8, SLC22A4, SLC26A4, SLC26A5, SLC4A1, SLC4A11, SLC52A2, SLC52A3, SLC6A13, SLITRK6, SMPX, SNAI2, SOX10, SOX2, SPINK5, STRC, SYNE4, TBC1D24, TBL1X, TBX1, TCF21, TCOF1, TECTA, TECTB, TFCP2, TIMM8A, TJP2, TMC1, TMEM126A, TMEM132E, TMIE, TMPRSS3, TMPRSS5, TNC, TPRN, TRIOBP, TSPEAR, TWNK, USH1C, USH1G, USH2A, WBP2, WFS1, WHRN, MT-RNR1, MT-TL1, MT-TS1*

Note. All the gene names were confirmed according to the updated records of HGNC (HUGO Gene Nomenclature Committee) database (https://www.genenames.org/, last accessed on 19 February 2023).

**Table 2 genes-14-00880-t002:** Clinical features of the study cohort.

	Study Cohort (N = 350)
Age, year	12.4 ± 13.8
Gender	
Female	171 (49)
Male	179 (51)
Age of onset	
Prelingual (<4 years)	173 (49)
Preschool (4–6 years)	70 (20)
Early school to teenage (6–18 years)	41 (12)
Young adult (18–40 years)	48 (14)
Late-onset (>40 years)	18 (5)
Severity of hearing loss	
Mild to moderate	234 (67)
Severe to profound	116 (33)

Note: Data are presented as mean ± standard deviation or number (%).

**Table 3 genes-14-00880-t003:** Clinical features of patients tested with different panels.

	30-Gene Panel (N = 271)	214-Gene Panel (N = 79)	*p*
Age, year	12.0 ± 12.3	13.8 ± 16.2	0.38
Gender			0.29
Female	137 (51)	34 (43)	
Male	134 (49)	45 (57)	
Age of onset			0.26
Prelingual (<4 years)	130 (48)	43 (54)	
Preschool (4–6 years)	57 (21)	13 (16)	
Early school to teenage (6–18 years)	36 (13)	5 (6)	
Young adult (18–40 years)	36 (13)	12 (15)	
Late-onset (>40 years)	12 (4)	6 (8)	
Severity of hearing loss			<0.001 ***
Mild to moderate	198 (73)	36 (46)	
Severe to profound	73 (27)	43 (54)	

Note: data are presented as mean ± standard deviation or number (%). *** *p* < 0.001.

**Table 4 genes-14-00880-t004:** Comparison of diagnostic rates between different targeted NGS examinations.

	30-Gene Panel (N = 271)	214-Gene Panel (N = 79)	*p*
Overall	138/271 (51)	44/79 (56)	0.56
Severity of hearing loss			
Mild to moderate	100/198 (51)	19/36 (53)	0.80
Severe to profound	38/73 (52)	25/43 (58)	0.53
Age of onset			
Prelingual (<4 years)	71/130 (55)	23/43 (53)	0.90
Preschool (4–6 years)	29/57 (51)	9/13 (69)	0.35
Early school to teenage (6–18 years)	17/36 (47)	3/5 (60)	0.66
Young adult (18–40 years)	19/36 (53)	4/12 (33)	0.32
Late-onset (>40 years)	2/12 (17)	5/6 (83)	0.01 *

Note: Data are presented as the number of patients with confirmed diagnosis/ the total number of patients (percentage); chi-square/Fisher’s exact test was used for categorical variables when appropriate. * *p* < 0.05.

**Table 5 genes-14-00880-t005:** Prognosis of hearing in patients with different genetic diagnoses.

	Stable	Progressive Hearing Loss or Profound Hearing Loss at Diagnosis	OR (CI) ^a^	*p ^a^*
Negative	42 (40)	62 (60)	[Ref]	[Ref]
Confirmed diagnosis				
*GJB2* variants	34 (59)	24 (41)	0.48 (0.25,0.92)	0.03 *
*SLC26A4* variants	5 (26)	14 (74)	1.90 (0.64, 5.66)	0.28
*MYO15A* variants	0 (0)	11 (100)	14.9 (0.85, 260.5) ^b^	0.007 **
*OTOF* variants	2 (33)	4 (67)	1.35 (0.24, 7.73)	1.00

Note: Data are presented as the number of patients (percentage). Data of patients with other genetic diagnoses were omitted owing to limited number of patients for comparison. * *p* < 0.05; ** *p* < 0.01. OR: odds ratio; CI: confidence interval. ^a^ The ORs, CI, and *p* values were calculated through comparisons with patients with negative genetic diagnosis. ^b^ The OR and CI were corrected for the zero-count cell.

## Data Availability

The data presented in this study are available on request from the corresponding author. The data are not publicly available due to ethical restrictions.

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
