# Peer review of "Revisiting Genetic Epidemiology with a Refined Targeted Gene Panel for Hereditary Hearing Impairment in the Taiwanese Population"

_genes, 2023, doi:10.3390/genes14040880_

Round 1
Reviewer 1 Report
The authors from Taipei, Taiwan report about their results and comparison of results from NGS of a 30 gene panel against the previous version with 214 genes panel. They tested 350 patients of Taiwanese origin with idiopathic hearing impairment of various degree and various age of onset, not only prelingual or early onset. The 30 gene panel was set from the previous results from the larger 214 gene panel. They found no significant difference in the diagnostic yields between the two gene panels – by both the panels about 50 % patients were clarified.
The paper is very well written and the study design is very precise, but there are several critical issues which should be improved and also discussed.
It is hardly possible or even impossible to seriously compare results from the two different gene panel if the patients groups were different – for the 30 genes panel different group of patients was tested than for the 214 gens panel.
Another important thing is, that the authors do not list the detected variants / mutations in any of the genes. It should be added at least for the relevant genes, which are mentioned in the charts and figures 1 and 2.
Which variants detected in the GJB2 genes classified the authors as pathogenic ?? Did they classify as pathogenic also the p.M34T and p.V37I ?? Was also the non coding exon 1 of the GJB2 gene sequenced ??
Why were the TMPRSS3, LOXHD1 and CDH23 genes, which are important in several other studies, not included in the simplified 30 gene panel ?
And what about the STRC gene deletion ?? Was it also tested – for example by CNV analysis ?? – Is it possible, that among the unclarified patients tested by the gene panels, there are some homozygotes for STRC gene deletion ?? Which is, in several populations the second most frequent cause of NSHL.
If the STRC gene deletions were not tested yet, it should be done – added.
In the reference list, which is quite long there are only few, if any paper reporting the NGS gene panel testing for hearing impairment. More such paper should be listed and discussed in the discussion to compare the results from other centers and population, for example this : https://doi.org/10.1111/cge.13839
Reviewer 2 Report
The manuscript is scientifically relevant, interesting, and well-written.
I have just a few issues to improve it.
The sentence 'The hearing prognosis of patients with negative genetic results varies and may decline over time' does not make sense to me since decline may occur with or without a diagnosis
The same for the sentence: 'Although highly variable, hearing deterioration is not uncommon in patients with negative genetic testing results, and appropriate counselling should be provided.' Please revise, including spelling. What is the message that you are willing to pass? Part of the negative results may be due to genes not covered by the panel or yet to be identified. This issue is well explained at the end of the discussion.
Also addressed at the end of the discussion, are the study limitations, I believe that excluding the cases with probable environmental etiology should also be discussed
